# Responses of Adult *Hypera rumicis* L. to Synthetic Plant Volatile Blends

**DOI:** 10.3390/molecules27196290

**Published:** 2022-09-23

**Authors:** Dariusz Piesik, Jan Bocianowski, Karol Kotwica, Grzegorz Lemańczyk, Magdalena Piesik, Veronika Ruzsanyi, Chris A. Mayhew

**Affiliations:** 1Department of Biology and Plant Protection, Faculty of Agriculture and Biotechnology, Bydgoszcz University of Science and Technology, 7 Prof. Kaliskiego Ave., 85-796 Bydgoszcz, Poland; 2Department of Mathematical and Statistical Methods, Poznań University of Life Sciences, 28 Wojska Polskiego, 60-637 Poznań, Poland; 3Department of Agronomy, Faculty of Agriculture and Biotechnology, Bydgoszcz University of Science and Technology, 7 Prof. Kaliskiego Ave., 85-796 Bydgoszcz, Poland; 4Oncology Center of F. Łukaszczyk in Bydgoszcz, 2 I. Romanowskiej St., 85-796 Bydgoszcz, Poland; 5Institute for Breath Research, University of Innsbruck and Tiroler Krebsforschungsinstitut (TKFI), Innrain 66, A-6020 Innsbruck, Austria

**Keywords:** curculionidae, *Hypera rumicis* L., VOCs, odors, orientation response

## Abstract

The behavioral responses of *Hypera rumicis* L. adults to varying blends of synthetic plant volatiles (SPVs) at various concentrations in lieu of single compounds are reported for the first time. For this study, *Rumex confertus* plants were treated with two blends of SPVs at different quantities that act as either attractants or repellents to insects. Blend 1 (B1) consisted of five green leaf volatiles (GLVs), namely (*Z*)-3-hexenal, (*E*)-2-hexenal, (*Z*)-3-hexenol, (*E*)-2-hexenol, and (*Z*)-3-hexen-1-yl acetate. Blend 2 (B2) contained six plant volatiles, namely (*Z*)-ocimene, linalool, benzyl acetate, methyl salicylate, β-caryophyllene, and (*E*)-β-farnesene. Each blend was made available in four different amounts of volatiles, corresponding to each compound being added to 50 µL of hexane in amounts of 1, 5, 25 and 125 ng. The effects of the two blends at the different concentrations on the insects were evaluated using a Y-tube olfactometer. Both sexes of the insects were found to be significantly repelled by the highest volatile levels of B1 and by two levels of B2 (25 and 125 ng). Females were also observed to be repelled using B2 with 5 ng of each volatile. Attraction was observed for both sexes only for B1 at the three lower volatile levels (1, 5 and 25 ng). In additional experiments, using only attractants, unmated females were found to be attracted to males, whereas mated females were only attracted to B1. Both unmated and mated males (previously observed *in copula*) were attracted only to females.

## 1. Introduction

Almost half of all insect species that attack plants are known herbivores, with crops being more susceptible than wild plants to attack [1]. A common plant herbivore found throughout Europe is *Hypera rumicis* (Linnaeus) (Coleoptera: Curculionidae), which is the subject of this paper. It is a pale buff colored weevil, of approximately 5 mm in length, with variable dark brown markings on the elytra. It feeds on docks and some other plants during late spring and summer, where the larva develops inside a woven spherical cage. A dock weevil (*H. rumicis*) is an herbivorous weevil and potential biocontrol agent, which can control the growth of *R. confertus* populations. The larvae and adults feed on the inflorescence stem and may destroy developing seeds. Moreover, when *H. rumicis* L. was added to curly dock (*Rumex crispus* L.), plants lost herbaceous green tissue, and in the longer term produced no seeds. In a natural field infestation, heavily infested plants were found to be lower in height and aerial biomass than lightly infested plants 2 to 7 m away, with no seeds being produced by the heavily infested plants [2]. Although it may control weeds, it can also destroy crops. Synthetic herbicide treatment has so far been the default method for controlling this pest. However, the number of active substances available for such use is shrinking [3,4]. This is because the widespread use of cost-effective chemicals has fostered insect resistance, as, for example, shown in *R. confertus* populations [5,6]. Generally, their long-term application has driven the development of resistance in insect pests to the substances contained in the insecticides [7,8]. Thus, high crop yields and the protection of crops from pathogens are no longer ensured.

Although plants are vulnerable to attack by organisms, they are not passive victims. To protect themselves actively, they have evolved an arsenal of chemical defenses, and specifically volatile emissions [9,10,11,12]. The emission of specific volatile organic compound (VOC) blends from plants varies extensively, depending on several factors, including the species, their developmental stage [13], propagation, defense [14], as well as local abiotic and biotic conditions [15,16,17,18]. The composition of VOCs induced by plants can be influenced by mechanical injury [19,20], herbivore feeding and development [21], and egg deposition [22]. The VOCs released by plants can also provide valuable information on the number of herbivores that are currently feeding on them [23].

Terpenes are the largest and most diversified class of secondary metabolites emitted by plants [24,25,26], and they can trigger a plant’s chemical defense mechanism [27]. The enemies in tritrophic interactions, such as parasitoids, can be attracted to a plant by the VOCs it emits in response to a herbivore attack or pathogen infection [28,29,30,31,32,33]. Host recognition may be achieved by recognition of species-characteristic combinations (blends) of plant volatiles [34,35,36,37,38,39].

Dock and sorrel (*Rumex*) populations are apt at habitat expansion and are hosts for *Hypera rumicis* L. [40]. The mossy sorrel (*Rumex confertus*-Polygonaceae) is an invasive plant that can spread rapidly; hence, it is considered a weed in crop fields for which biological control agents are sought. There have been some examples of successful biocontrol agent establishment on *Rumex* spp., where, in Western Australia, *R. pulcher* L. were greatly reduced. Moreover, *Rumex* species are used as food plants by the larvae and/or adults of many Lepidoptera and Coleoptera species, offering a large pool of possible biological control agents to explore [41,42,43].

An insect’s response to plant VOCs provides information on the types of volatiles and concentration levels over which herbivore attraction or repulsion can occur [44,45,46,47]. Experiments in which insects are exposed to plant volatiles, either individually or in combination, have revealed that stronger behavioral responses can be obtained by using appropriate blends rather than single compounds [48,49].

A major aim of this study is to further our understanding of insect’s behavioral responses (attraction or repulsion), specifically *Hypera rumicis* L., following a plant’s exposure to two different blends of plant volatiles, with each blend consisting of four different quantities of the VOCs, rather than concentrating on the effects of individual volatile compounds. The importance of studies such as this one is that they can potentially lead to the development of plant volatile blends for the treatment of crops to limit pest attack, thereby leading to higher crop yields with lower risks to the environment, hence being more acceptable to current trends in having organic products.

Here, we report for the first time the effects of varying blends of plant-induced VOCs on *Hypera rumicis* L. that infest *R*. *confertus*. Behavioral responses to the blends of both male and female insects and of unmated or mated insects of both sexes are also reported.

## 2. Results

The MANOVA indicated a significant difference among the concentrations for all of the 11 VOCs jointly (Wilk’s = 0.001117; F_3;36_ = 21.19; *p* < 0.0001). The ANOVA for (*Z*)-3-HAL (F_3;36_ = 265.14), (*E*)-2-HAL (F_3;36_ = 132.38), (*Z*)-3-HOL (F_3;36_ = 104.85), (*E*)-2-HOL (F_3;36_ = 70.76), (*Z*)-3-HAC (F_3;36_ = 277.05), (*Z*)-OCI (F_3;36_ = 181.98), LIN (F_3;36_ = 285.63), BAC (F_3;36_ = 62.10), MAT (F_3;36_ = 108.74), β-CAR (F_3;36_ = 286.38), and (*E*)-β-FAR (F_3;36_ = 819.13) confirmed the variability of the tested concentrations at a significance level of α = 0.001 (Table 1).

The mean values for the observed VOCs are presented in the radar chart (see Figure 1). These results indicate high variability among the tested concentrations, for which significant differences were found in terms of all of the analyzed VOCs. All pairs of observed VOCs had statistically significant differences at the 0.001 level. All correlation coefficients were positive. The greatest variation among all of the 11 VOCs was found for the 125 ng and control, with a Mahalanobis distance of 59.07. The greatest similarity was found for hexane and the control (0.29).

Of the two blends of SPVs tested at four doses (1, 5, 25, and 125 ng of each volatile in 50 µL of hexane), both sexes were significantly repelled by B1 containing the highest level of volatiles (125 ng) and two consecutive doses of B2 (25 and 125 ng) (Table 2). Moreover, females were also repelled by B2 containing VOCs at a level of 5 ng/VOC.

Attraction was observed only for B1 at three levels (1, 5, and 25 ng/VOC) (see Table 3). When only the attractant doses were taken into consideration, we found that unmated females were attracted to males (Table 3), and mated ones were only attracted to B1 at 5 and 25 ng·min^−1^ (Table 4). Both unmated and mated males were attracted only to females at 5 and 25 ng·min^−1^ not to B1 (Table 3 and Table 4). It was found in the research that unmated individuals of both sexes react to the tested mixture and individuals of the opposite sex at a similar level (5 ng·min^−1^–4.05* (f/a/m)^3^ and 4.05* (m/a/f)^4^; 25 ng·min^−1^–6.05* (f/a/m)^3^ and 8.45** (m/a/f)^4^). A different situation was observed for mated insects. The males did not change much the intensity of selection of the mixture or the individuals of the opposite sex (5 ng·min^−1^–4.05* (m/a/f)^4^; 25 ng·min^−1^ 6.05* (m/a/f)^4^). However, mated females were more interested in the volatile compounds than males (5 ng·min^−1^–8.45** (f/a)^2^; 25 ng·min^−1^–11.25*** (f/a)^2^). The intensity of selection increased significantly. This may suggest that mated females are more interested in laying eggs, which is important for the species survival.

## 3. Discussion

Various studies have found that common plant volatiles that are emitted by the damaged leaves of a plant include five GLVs-(*Z*)-3-HAL, (*E*)-2-HAL, (*Z*)-3-HOL, (*E*)-2-HOL, (*Z*)-3-HAC], and six VOCs-(*Z*)-OCI, LIN, BAC, MAT, β-CAR, and (*E*)-β-FAR. In comparison to damaged leaves, a plant’s undamaged leaves constitutively emit smaller quantities of volatiles [1]. It is well known that plant volatiles can repel insects or can act as cues to predator insects, guiding them to infected plants [48,50]. Externally manipulating these signals could potentially improve their effectiveness in the attraction of predatory natural enemies for use as more effective biological control agents [51,52]. To date, the majority of investigations have focused on testing single volatile compounds [53]. Yet, we have found that VOC blends containing appropriate amounts of the VOCs are of more importance to the behavior of insects (herbivores and predators alike).

In this study, both sexes of *H. rumicis* responded negatively to B1 and B2 containing the highest volatile levels. B1 was found to be attractive to females for blends containing volatiles at levels of 1 up to 25 ng/VOC. Moreover, different responses were observed for unmated or mated insects of both sexes to B1 or unmated/mated individuals. This is in good agreement to Webster et al. [54], who found that insects usually rely on between 3 to 10 compounds for host plant recognition. Moreover, Ali and Wright [55] demonstrated in a Y-tube olfactometer that female wasps were significantly attracted to two volatiles among the seven compounds tested. Piesik et al. [30] also found that *Cephus cinctus* Norton females were attracted to some doses of (*Z*)-3-HAC and (*Z*)-3-HOL. However, they were repelled by the highest tested doses (8400 ng·h^−1^) of (*Z*)-3-HAC, close to the deterrent concentration of GLVs to *Gastrophysa viridula* and *Gastrophysa polygoni.* Moreover, in maize, Fusarium infection induced VOCs, and adult cereal leaf beetles (*Oulema melanopus* Linnaeus) were attracted to synthetic components at doses of 7500 ng·h^−1^ for the two GLVs, (*Z*)-3-HAL and (*Z*)-3-HAC, and two terpenes, LIN and β-CAR, as well as at lower doses of 60 ng·h^−1^ for both GLVs and 300 ng·h^−1^ for LIN. Furthermore, Piesik et al. [30] found that even the lowest concentrations tested (1 ng·min^−1^) repelled females of *G. viridula*. These results suggest that the blend of volatiles tested was sufficient for achieving repellence. In other cases, VOCs at low concentrations and at high concentrations may act as attractants or as repellents, respectively. By contrast, Carroll et al. [56] showed that *Spodoptera frugiperda* (J. E. Smith) moths were more attracted to injured maize than to uninjured plants for a wide range of LIN concentrations.

In our experiments, a synthetic blend of attractants versus unmated/mated individual insects were tested for the first time. No similar experiments have been conducted elsewhere. Unmated females searched for males, whereas mated ones responded positively to a blend of volatiles. This may suggest that they were looking for a host to provide suitable food for themselves and their offspring. We have shown that insects respond to volatiles in different ways that depend not only on the volatile and/or its concentration, but also on priority. This is in good agreement with Beyaert et al. [57], who reported that olfactory orientation by insects might be guided by specific volatile blends released from sites where resources are present. This is also in good agreement with Davidson-Lowe and Ali [58], who found that herbivores demonstrate contrasting induction of plant volatiles and behavioral responses. This aids us to improve our understanding of the ecological functions and community dynamics of plant plasticity and interactions with a variety of herbivores. All of this points to the potential development of eco-friendly strategies, for which aromatic plants may offer a promising alternative due to their repellent properties [59,60,61,62,63,64,65].

Unmated and mated females and males were tested against the opposite sex and a mixture of volatile attractants (B1). The reaction of the individuals to the opposite sex before and after mating has been shown. This type of study has not been carried out before. Importantly, it highlights a change in female interest depending on fertilization status.

## 4. Materials and Methods

The experiments reported in this paper were conducted in 2017 at the UTP University of Science and Technology in Bydgoszcz, Poland, at the Department of Entomology and Molecular Phytopathology (recently renamed the Bydgoszcz University of Science and Technology, the Department of Biology and Plant Protection, Laboratory of Entomology) and at the Nicolaus Copernicus University in Toruń, Poland, at the Department of Environmental Chemistry and Bioanalytics. The experimental workflow method adopted in this study consisted of three phases.

The first phase was a plant’s response to blends and doses of VOCs, where one leaf of the plant was treated with one of two blends of VOCs at two doses, namely blends containing 5 ng and 125 of each VOC (see below).

Using a Y-tube olfactometer, the second phase involved investigations of *H*. *rumicis* adult behavior to the two blends containing varying VOC amounts.

The third, and final phase, consisted of studies for which a synthetic blend of attractants (at three volatile levels) was compared to insect attraction of the opposite sex.

### 4.1. Plants

*R. confertus* plants were planted and cultivated in a greenhouse at ambient humidity with supplemental light involving a photoperiod of 16 h (light) and 8 h (dark) at temperatures of 22 ± 2 °C and 18 ± 2 °C, respectively. Thirty of these plants were dug up and transplanted into pots containing sterilized soil, with one plant per pot, at the Plant Growth Centre. Two months after transplantation, the plants were randomly assigned to groups being exposed to synthetic blends of VOCs or hexane, or to a control group.

### 4.2. Application of VOC Blends

The compounds known to appear in defense reactions in cereal crops and also in *R. confertus* (previous experiments, [59,66]) were selected for the experiments, namely: (*Z*)-3-HAL [(*Z*)-3-hexenal], (*E*)-2-HAL [(*E*)-2-hexenal)], (*Z*)-3-HOL [(*Z*)-3-hexenol], (*E*)-2-HOL [(*E*)-hexenol], (*Z*)-3-HAC [(*Z*)-3-hexen-1-yl acetate], (*Z*)-OCI [(*Z*)-β-ocimene], LIN [linalool], BAC [benzyl acetate], MAT [methyl salicylate], β-CAR [β-caryophyllene], and (*E*)-β-FAR [(*E*)-β-farnesene]. For convenience, in the following text, the acronyms for the volatiles will be mainly used. It is known that the beetles under investigation are capable of sensing these compounds.

Insects respond to quite small emission rates of volatiles from plants (e.g., in the range of 1–100 ng·min^−1^). Thus, we decided to investigate the emission rates (controlled by an air flow meter) between 1 and 125 ng·min^−1^.

To create a specific blend, a quantity (1, 5, 25 or 125 ng) of each selected VOC (95% purity; Sigma-Aldrich, Poznań, Poland) was added to 50 µL of hexane.

Before any exposure to the VOCs occurred, selected plants were transferred from the greenhouse to the laboratory. A given VOC blend containing known amounts of volatiles was poured onto one quarter of a 55 mm disk of filter paper, which was folded and placed into a microcentrifuge tube. A 27.5-gauge syringe needle was pushed through each microcentrifuge tube cap immediately before the first filter was added. One leaf from each *R*. *confertus* plant was placed in a plastic cylinder, along with a microcentrifuge tube. One hole was made on the top and at the bottom of the plastic cylinder (20 cm height and 6 cm diameter) to surround the inserted leaf petiole. The filter paper was replaced with a new one every hour. Plants were exposed to a given VOC blend containing known quantities of volatiles (1, 5, 25 or 125 ng of each volatile) for a total of 6 h.

### 4.3. Volatile Collection System

The volatiles being emitted from four single enclosed leaves on *R. confertus* were collected simultaneously into Nalophan bags. These bags are odor- and taste-free consisting of a plastic film that is resistant to temperatures ranging from 60 °C to 220 °C. Each sampling period lasted for 3 h (from 10:00 to 13:00); this procedure was carried out over 8 days (providing 32 samples for the measurements), in which treated plants were randomly assigned to collection days (light/dark and temperature conditions the same as for plants). A volatile collector trap (glass tube with an outer diameter of 6.35 mm and a height of 76 mm; Analytical Research Systems, Inc., Gainesville, FL, USA) containing 30 mg of Super-Q (Alltech Associates, Inc., Deerfield, Illinois, USA) adsorbent was inserted into each of four Tygon tubes (which connected the airflow meter and the collector trap). Purified, humidified air was delivered at a rate of 1.0 L·min^−1^ over the plants, and a vacuum pump sucked 20% less (0.8 L·min^−1^) to avoid collecting volatiles from any gap in the system.

### 4.4. Analytical Methods

Volatiles were eluted from the Super-Q collection trap with 225 µL of hexane containing 25 ng of decane as an internal standard. Volatiles were analyzed by gas chromatography-mass spectrometry (GC-MS). A GC Perkin Elmer AutoSystem XL was fitted with a 30 m DB-5MS capillary column (inner diameter of 0.25 mm, and film thickness 0.25 µm; Restek, Bellefonte, FL, USA). The temperature program increased the temperature of the chromatography oven from 40 °C to 200 °C at a rate of 5 °C·min^−1^. Plant volatile identification was verified using authentic standards (Sigma-Aldrich, Poznań, Poland) with the same GC retention times and mass spectra. Peaks were integrated directly from the GC-chromatogram, and they were compared to GC-MS library and the internal standard peak to determine the concentration. However, it should be noted that both the Z and E isomers of β-ocimene were present in the standard solution.

### 4.5. Insects

Adult *H*. *rumicis* weevils (males and females identified *in copula*) were collected in meadows by the Vistula River near to Bydgoszcz, at the coordinates 53°9’7.039” N, 18°11’3.135” E. They were reared on potted *R*. *confertus* plants under a normal light regime (16 h day: 8 h night) at a temperature of 22 ± 2 °C and at a relative humidity of 60 ± 5%. In the experiments, only newly emerged adults were used.

### 4.6. Synthetic Chemicals

Two blends, each having four mixtures with four different levels of synthetic plant volatiles added, were made-up for this study. The chemicals used were obtained from Sigma-Aldrich (Chemical Co. Inc., Poznań, Poland), with stated purities of 85–99%. The selection of the volatiles was based on emissions from various cereal greens that had been induced by biotic stress [30,31,66]. Blend 1 consisted of five GLVs: (*Z*)-3-HAL, (*E*)-2-HAL, (*Z*)-3-HOL, (*E*)-2-HOL, and (*Z*)-3-HAC. Blend 2 (B2) was made up of six volatiles: (*Z*)-OCI, LIN, BAC, MAT, β-CAR, and (*E*)-β-FAR.

### 4.7. Y-Tube Olfactometer

The Y-tube system used was similar to that described by Piesik et al. [50]. In brief, the interior angle of the ‘Y’ was 120°, and the diverging arms extended for 4 cm in each direction before becoming parallel for their final 10 cm, and then terminated in a female ground-glass joint at the end of each arm. In the first experiment to determine response, purified, humidified air with a test blend and purified, humidified air with hexane only, were delivered to both arms of the Y-tube. In the second experiment, to determine the response of unmated or mated female or male individuals to attractant blends, purified humidified air with a test concentration of the blend and purified, humidified air with hexane only, were delivered to both arms of the Y-tube. The air passed through a set of two gas wash bottles: one for capturing the insects who moved into one of the ports, and the other for loading the air with the test volatiles. The system was washed and rotated to limit chemical residues from a previous bioassay. The Y-tube and blend were changed after every single run. For screening the behavioral activity, four different amounts of volatiles for each of the two blends (1, 5, 25, and 125 ng of each volatile) were used, and compared to their behavior in the absence of any volatile blend, i.e., using hexane solvent alone.

Blends containing various amounts of the volatiles (each volatile contributing 1, 5, 25 or 125 ng) were individually placed into one arm of the Y-tube olfactometer. Hexane was introduced into the second arm of the olfactometer. Moreover, in the Y-tube olfactometer, B1, with three different quantities of VOCs, which have individually been previously found to be attractants, (see Table 2, where mixture B1 showed a positive insect response at low concentrations) were finally tested against the non-mated/mated female or male individuals.

The beetles were introduced into the system by lifting the glass cover, or by placing the insects into the central aperture using the introduction trap (light conditions). For each blend containing a known quantity of volatiles or for the product of unmated/mated female or male individuals, another 20 adult *H*. *rumicis* individuals of each sex were tested (always separately). The tests were run for 5 min each until a weevil selected one of the arms; this means that a given weevil was allowed a maximum of 5 min to choose one of the two arms. The behavior of the weevils in the olfactometer was recorded by direct observation during the day (from 10 am to 1 pm). Not surprisingly, non-responsive insects were also observed.

### 4.8. Release and Calibration of Odors

Manipulation and control of the desired rates of release of the synthetic volatiles for bioassays were achieved by varying the amount of material applied to the filter paper substrate in the microcentrifuge tube and by varying the number of holes in the cap of the microcentrifuge tube. One or more holes were made in the cap to facilitate the release of the volatiles, using either a standard pushpin or a 27.5-gauge syringe. The microcentrifuge cap was left open for 1 min to allow the hexane to evaporate. The release rates were quantified and calibrated using the volatile collection system and the GC-MS.

### 4.9. Statistical Analyses for VOC Collection

The normality of the distributions of the 11 VOCs under investigation were tested using Shapiro–Wilk’s normality test [67]. Multivariate analysis of variance (MANOVA) was used to test multivariate differences between concentrations of all the 11 VOCs jointly. MANOVA was performed using GenStat version 18 software and the equation Y = XT + E, where Y is the (*n* × *p*)-dimensional matrix of observations, *n* is the number of all observations, *p* is the number of VOCs (in this study *p* = 11), X is the (*n* × *k*)-dimensional matrix of design, *k* is the number of treatments (in this study, *k* = 4), T is the (*k* × *p*)-dimensional matrix of unknown effects, and E is the (*n* × *p*)-dimensional matrix of residuals. Next, we tested differences between concentrations for particular VOCs, independently. One-way analysis of variance (ANOVA) was conducted to determine the effects of the concentrations on the variability of (*Z*)-3-HAL, (*E*)-2-HAL, (*Z*)-3-HOL, (*E*)-2-HOL, (*Z*)-3-HAC, (*Z*)-OCI, LIN, BAC, MAT, β-CAR, and (*E*)-β-FAR. The mean values for the VOCs were calculated. Tukey’s honest least significant differences (HSDs) were calculated for the individual VOCs. The relationships between observed VOCs were assessed based on Pearson’s correlation. Mahalanobis distances [68] were suggested to measure the concentration distance for poly-VOC [69], whose significance was verified using the critical value D_α_, the least significant distance [70,71]. Mahalanobis distances were calculated for all concentrations. All of the analyses were conducted using the statistical software package GenStat, 18th edition.

### 4.10. Data Analyses for Y-Tube Olfactometer

The χ^2^ test with the Yates correction for small samples was conducted to determine goodness of fit in order to assess whether the weevils in the Y-tube olfactometer were attracted to, were repelled by, or had no response to the odor source (VOC blend versus hexane solvent or attractive VOC blend vs. non-mated/mated individuals) at each exposure concentration. The beetle counts observed did not significantly deviate from the expected ratio of 10:10 (the arm with the hexane solvent only versus the arm with the VOC blend or the arm with the VOC blend versus unmated/mated individuals). After 6 h from the mating event, the beetles were tested.

## 5. Conclusions

*R. confertus* plants were treated with common plant volatile attractants and repellents consisting of one of two blends, with each blend being prepared in four different volatile quantities (1, 5, 25 and 125 ng/VOC), which were evaluated using a Y-tube olfactometer. To create a specific blend, a quantity (1, 5, 25 or 125 ng) of each selected VOC was added to hexane. Blend 1 consisted of five GLVs: (*Z*)-3-hexenal, (*E*)-2-hexenal, (*Z*)-3-hexenol, (*E*)-2-hexenol, and (*Z*)-3-hexen-1-yl acetate. Blend 2 consisted of (*Z*)-ocimene, linalool, benzyl acetate, methyl salicylate, β-caryophyllene, and (*E*)-β-farnesene. Both sexes of the insect were found to be significantly repelled by the highest containing levels of volatiles for B1 and for two consecutive doses of B2. Females were found to be repelled by B2 at the 5 ng level. Attraction for both sexes was observed only for B1. Unmated females were attracted to males, and mated ones were only attracted to B1. Both unmated and mated males were attracted only to females.

## Figures and Tables

**Figure 1 molecules-27-06290-f001:**
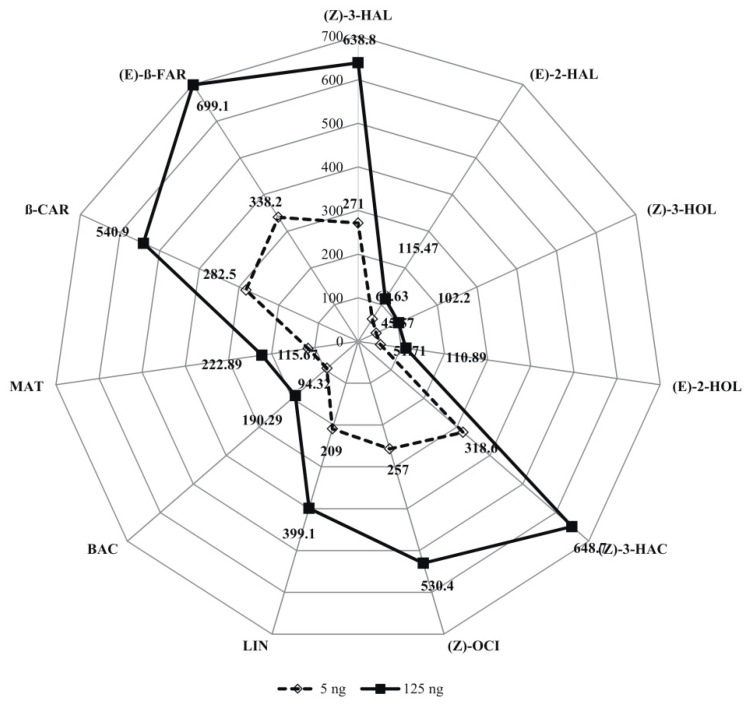
Radar chart of the mean values (in ng·h^−1^) of VOCs for four concentrations; HSD_0.001_ values for VOCs are as follows: (*Z*)-3-HAL, 93.72; (*E*)-2-HAL, 24.36; (*Z*)-3-HOL, 23.86; (*E*)-2-HOL, 31.51; (*Z*)-3-HAC, 93.88; (*Z*)-OCI, 94.79; LIN, 57.32; BAC, 58.00; MAT, 51.76; β-CAR, 77.57; and (*E*)-β-FAR, 58.8.

**Table 1 molecules-27-06290-t001:** Mean squares from one-way analysis of variance for the observed VOCs (where one leaf of the plant was treated with one of two blends of VOCs (B1 and B2) at two doses (5 and 125 ng·min^−1^).

Source of Variation	Concentrations	Residual
The number of degrees of freedom	3	36
(Z)-3-HAL	907,362 ***	3422
(E)-2-HAL	30,606.3 ***	231.2
(Z)-3-HOL	23,262.1 ***	221.9
(E)-2-HOL	27,375 ***	386.9
(Z)-3-HAC	951,397 ***	3434
(Z)-OCI	637,135 ***	3501
LIN	365,700 ***	1280
BAC	81,395 ***	1311
MAT	113,526 ***	1044
β-CAR	671,452 ***	2345
(E)-β-FAR	1,104,779 ***	1349

*** *p* < 0.001.

**Table 2 molecules-27-06290-t002:** Effects of rates of dose (ng·min^−1^) for synthetic blend 1 (B1), consisting of five GLVs [(*Z*)-3-HAL + (*E*)-2-HAL + (*Z*)-3-HOL + (*E*)-2-HOL + (*Z*)-3-HAC], and synthetic blend 2 (B2) consisting of six VOCs [(*Z*)-OCI + LIN + BAC + MAT + β-CAR + (*E*)-β-FAR] on the number of unmated *H. rumicis* adult females and males choosing to enter a Y-tube arm containing the blend or the Y-tube arm containing purified, humidified air and hexane solvent (no volatiles).

			No. of Females			No. of Males		
Name of mixed compounds	Rep.	ng·min^−1^	+ ^(4)^	– ^(5)^	χ^2 (1)^	+ ^(4)^	– ^(5)^	χ^2 (1)^
	control	0	12	8	0.45 ns	7	13	1.25 ns
(*Z*)-3-HAL								
+ (*E*)-2-HAL	1	1	15	5	4.05* (a) ^(3)^	8	12	0.45 ns
+ (*Z*)-3-HOL	2	5	16	4	6.05* (a) ^(3)^	11	9	0.05 ns
+ (*E*)-2-HOL	3	25	15	5	4.05* (a) ^(3)^	8	12	0.45 ns
+ (*Z*)-3-HAC	4	125	3	17	8.45** (r) ^(2)^	4	16	6.05* (r) ^(2)^
	control	0	10	10	0.05 ns	7	13	1.25 ns
(*Z*)-OCI								
+ LIN	1	1	11	9	0.05 ns	6	14	2.45 ns
+ BAC	2	5	3	17	8.45** (r) ^(2)^	9	11	0.05 ns
+ MAT	3	25	4	16	6.05* (r) ^(2)^	5	15	4.05* (r) ^(2)^
+ β-CAR	4	125	3	17	8.45** (r) ^(2)^	3	17	8.45** (r) ^(2)^
+ (*E*)-β-FAR								

Legend: (1) level of significance (ns–not significant), (* *p* < 0.05), (** *p* < 0.01), (2) r—repellent, (3) a—attractant, (4) + Y-tube arm with tested amount of the compound, volatile diluted in hexane emitted from filter paper, (5) – Y-tube arm only with hexane emitted from filter paper.

**Table 3 molecules-27-06290-t003:** Effects of rates of dose (ng·min^−1^) for synthetic blend 1 (B1), consisting of five GLVs [(*Z*)-3-HAL + (*E*)-2-HAL + (*Z*)-3-HOL + (*E*)-2-HOL + (*Z*)-3-HAC] (three tested doses—only attractants identified in Table 2) on the number of unmated *H. rumicis* adult females and males choosing to enter a Y-tube arm containing the volatile blend or the Y-tube arm containing female or male individuals.

			No. of Females			No. of Males		
	Rep.	ng·min^−1 (7)^	+ ^(5)^	– ^(6)^	χ^2 (1)^	+ ^(5)^	– ^(6)^	χ^2 (1)^
			(B1)	(M)		(B1)	(F)	
	1	1	10	10	0.05 ns	8	12	0.45 ns
Blend/*H. rumicis*	2	5	5	15	4.05* (f/a/m) ^(3)^	5	15	4.05* (m/a/f) ^(4)^
	3	25	4	16	6.05* (f/a/m) ^(3)^	3	17	8.45** (m/a/f) ^(4)^

Legend: (1) level of significance (ns–not significant), (* *p* < 0.05), (** *p* < 0.01), (3) f/a/m—females attracted to male, (4) m/a/f—males attracted to female, (5) + Y-tube arm with tested amount of the compounds, volatile diluted in hexane emitted from filter paper, (6) – Y-tube arm only with female or male individual, (7) one arm with tested blend (from 1 to 25 ng.min^‒1^) and second one with single insect (from 1 to 25 ng.min^‒1^, purified, humidified air), B1—blend 1, M—male, F—female.

**Table 4 molecules-27-06290-t004:** Effects of rates of dose (ng·min^−1^) for synthetic blend 1 (B1), consisting of five GLVs [(*Z*)-3-HAL + (*E*)-2-HAL + (*Z*)-3-HOL + (*E*)-2-HOL + (*Z*)-3-HAC] (three tested doses—only attractants identified in Table 2) on the number of mated *H. rumicis* adult females and males choosing to enter a Y-tube arm containing the volatile blend or the Y-tube arm containing only female or male individuals.

			No. of Females			No. of Males		
	Rep.	ng·min^−1 (7)^	+ ^(5)^	– ^(6)^	χ^2 (1)^	+ ^(5)^	– ^(6)^	χ^2 (1)^
			(B1)	(M)		(B1)	(F)	
	1	1	14	6	2.45 ns	11	9	0.05 ns
Blend/*H. rumicis*	2	5	17	3	8.45** (f/a) ^(2)^	5	15	4.05* (m/a/f) ^(4)^
	3	25	18	2	11.25*** (f/a) ^(2)^	4	16	6.05* (m/a/f) ^(4)^

Legend: (1) level of significance (ns–not significant), (* *p* < 0.05), (** *p* < 0.01), (*** *p* < 0.001), (2) f/a—females attracted to B1, (4) m/a/f—males attracted to female, (5) + Y-tube arm with tested amount of the compounds, volatile diluted in hexane emitted from filter paper, (6) – Y-tube arm only with female or male individual, (7) one arm with tested blend (from 1 to 25 ng.min^‒1^) and second one with single insect (from 1 to 25 ng.min^‒1^, purified, humidified air), B1—blend 1, M—male, F—female.

## Data Availability

Not applicable.

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
