# Peer review of "Responses of Adult Hypera rumicis L. to Synthetic Plant Volatile Blends"

_molecules, 2022, doi:10.3390/molecules27196290_

Round 1
Reviewer 1 Report
Review of manuscript Responses of adult Hypera rumicis L. to synthetic plant volatile blends
The tested Insect is not described. Not clear why this insect is tested. Which order and family is it? Which weed does it control? Pleas add more information.
The selection of compounds and blends is not justified and explained well. There are references on emission from plants but not about the insect eventual sensing of the compounds. Do we know if the beetle is capable of sensing these compounds?
In line 201 it is said that compounds were previously attractant. This needs more clarification. Which study?
The authors have tested mated and unmated individuals. Yet there is no explanation about why they did it and what the explanation of the results are very shallow and need more thoughts.
Attraction to the opposite sex were also tested. Here, attraction to the same test would have been needed to be able to compare and say anything about the importance.
There are many questions about the relevance of the study that need explanation. If the authors could explain more about the relation between compound emission from Rumex and the tested compounds and the relation between the insect and the compounds, the study would be more comprehensive.
Author Response
Thank you for the review. All of the suggestions have been included and attached to the new version of the manuscript.

Reviewer 2 Report
The behavioral responses of Hypera rumicis adults (unmated/mated individual insects) to varying blends of synthetic plant volatilesat various concentration of volatiles. It's a breakthrough that use various concentration of volatiles instead of single-origin compounds. I found the study of interest and a good contribution to the knowledge of bioecology of pests. The authors have graphed and presented their results clearly. The methods used are appropriate for the objectives of the work and, in general, well depicted. However, several parts of the manuscript need to be presented better and in more details. A careful Language editing is also needed. I provide specific comments below.
Line 46-47, you means use Synthetic herbicide to control insect pest Hypera rumicis?
Line117, why choose these compounds? Is there references or for some reasons?
Line123, how to control the emission rates?
Table1 it is better to let us know if the VOCs was collected from normal plants or treated plants? Is this information was described in section2.9?
Line408, “which were evaluated using a using a Y-tube olfac-408 tometer” there are two “using a ” in this sentence.
To create a specific blend, a quantity (1, 5, 25 or 125 ng) of each selected VOC was added to hexane.
You mean, each VOC has the same content in the blend? From the results, concentrations of VOCs were different. Maybe it is better to simulate the proportion of the natural VOCs. This is not to diminish the data gathered in this study, they are of value.
Author Response

(The authors gave the same response as above.)
